# The Role of Peroxisome Proliferator-Activated Receptors in Preeclampsia

**DOI:** 10.3390/cells12040647

**Published:** 2023-02-17

**Authors:** Iason Psilopatis, Kleio Vrettou, Florian Nima Fleckenstein, Stamatios Theocharis

**Affiliations:** 1Department of Diagnostic and Interventional Radiology, Charité—Universitätsmedizin Berlin, Corporate Member of Freie Universität Berlin and Humboldt—Universität zu Berlin, Augustenburger Platz 1, 13353 Berlin, Germany; 2First Department of Pathology, Medical School, National and Kapodistrian University of Athens, 75 Mikras Asias Street, Bld 10, Goudi, 11527 Athens, Greece; 3BIH Charité Clinician Scientist Program, Berlin Institute of Health at Charité—Universitätsmedizin Berlin, BIH Biomedical Innovation Academy, 10117 Berlin, Germany

**Keywords:** peroxisome proliferator-activated receptor (PPAR), preeclampsia

## Abstract

Preeclampsia is a common pregnancy-related hypertensive disorder. Often presenting as preexisting or new-onset hypertension complicated by proteinuria and/or end-organ dysfunction, preeclampsia significantly correlates with maternal and perinatal morbidity and mortality. Peroxisome proliferator-activated receptors (PPARs) are nuclear receptor proteins that regulate gene expression. In order to investigate the role of PPARs in the pathophysiology of preeclampsia, we conducted a literature review using the MEDLINE and LIVIVO databases. The search terms “peroxisome proliferator-activated receptor”, “PPAR”, and “preeclampsia” were employed and we were able to identify 35 relevant studies published between 2002 and 2022. Different study groups reached contradictory conclusions in terms of PPAR expression in preeclamptic placentae. Interestingly, PPARγ agonists alone, or in combination with well-established pharmaceutical agents, were determined to represent novel, potent anti-preeclamptic treatment alternatives. In conclusion, PPARs seem to play a significant role in preeclampsia.

## 1. Introduction

Preeclampsia is a form of pregnancy disorder characterized by the onset of hypertension, proteinuria and/or end-organ dysfunction after 20 weeks’ gestation [1]. Affecting circa 2–8% of pregnancies globally, preeclampsia represents one of the most common causes of both maternal and perinatal morbidity and mortality [2,3]. Clinically, preeclampsia typically presents with new-onset hypertension and proteinuria at ≥34 weeks of gestation, with severe clinical features ranging from persistent headache, nausea, or vomiting to visual abnormalities, abdominal discomfort, and altered mental status [2,4]. Diagnostic evaluation includes, apart from a detailed history and physical exam, a urine analysis, complete blood count, a complete metabolic panel, as well as close control of renal retention parameters [5]. According to level-A guideline recommendations published by The American College of Obstetricians and Gynecologists (ACOG), women with preeclampsia should receive magnesium sulfate as prophylaxis and/or treatment of seizures, whereas (preterm) delivery is suggested for patients with preeclampsia without severe clinical symptoms at or beyond 37 0/7 weeks of gestation [3]. To date, the exact pathomechanisms of preeclampsia still remain unknown, with various theories proposing abnormal placentation, immunologic factors, inflammation, genetics, as well as acquired risk factors as the most relevant causes [6].

Peroxisome proliferator-activated receptors (PPARs) are fatty acid-activated nuclear receptors that comprise of three isoforms with distinct metabolic regulatory activities, tissue distribution, and ligand-binding assets: α, β/δ, and γ [7,8,9]. PPARs play a regulatory role in fatty acid disposition and metabolism, energy homeostasis, cellular biology functions, cell differentiation, as well as immunity pathways [10,11]. More precisely, PPARα and PPARβ/δ are mainly involved in energy combustion, whereas PPARγ enhances adipogenesis [12]. Importantly, PPAR isotypes have been described to be expressed in the placenta and to influence placental development, homeostasis, endometrium decidualization, fetal implantation, trophoblast physiological conditions, and oxidative pathways [13,14,15].

Given the involvement of PPARs in various placental functions, a literature review was conducted in order to investigate the their role in the pathophysiology of preeclampsia. The literature review was conducted using the MEDLINE and LIVIVO databases. Solely original research articles and scientific abstracts written in the English language that explicitly reported on the role of PPARs in the preeclamptic pathogenesis were included in the data analysis. Studies emphasizing the involvement of PPARs in uncomplicated human pregnancy or obstetric and gynecologic disorders other than preeclampsia (e.g., polycystic ovarian syndrome, endometriosis, intrauterine growth restriction, gestational diabetes) were excluded. The search terms “peroxisome proliferator-activated receptor”, “PPAR”, and “preeclampsia” were employed, and we were able to identify a total of 94 articles published between 2000 and 2022, after the exclusion of duplicates. A total of 49 were discarded in the initial selection process after abstract review. The full texts of the remaining 45 publications were evaluated, and after detailed analysis, a total of 35 relevant studies published between 2002 and 2022 that met the inclusion criteria were selected for the literature review. Figure 1 presents an overview of the aforementioned selection process.

## 2. PPAR Structure and Mechanism of Action

PPARs consist of four functional (A/B, C, D and E/F) [16] and two binding domains (a DNA binding domain in the *N*-terminus and a ligand binding domain in the C-terminus) [17]. By binding to specific DNA response elements within promoters as heterodimers with the retinoid X receptors (RXRs), PPARs represent ligand-regulated transcription factors which promote or inhibit the expression of the target gene (Figure 2) [18]. These DNA regions of PPAR-responsive genes are termed peroxisome proliferator hormone response elements (PPREs) [18]. Transcription of PPAR-regulated genes is amplified by co-activators that either remodel chromatin structure by the intrinsic histone acetyltransferase, methyltransferase or helicase activities (i.e., Steroid Receptor Co-activator (SRC), CREB-binding protein (CBP)/p300, PPAR-interacting protein (PRIP) with methyltransferase domain (PIMT)/Nuclear receptor co-activator 6 (NCoA6)-interacting protein (NCoA6IP), PPARα-receptor interacting co-factor (PRIC), co-activator-associated arginine methyl-transferase 1 (CARM1), SWItch/Sucrose Non-Fermentable (SWI/SNF)), or do not possess any known enzymatic functions but recruit multisubunit protein transcriptional complexes (i.e., Mediator complex subunit 1 (MED1), PPARγ co-activator 1 (PGC1), PRIP/NCoA6, BRG1/BRM-associated factor 60 (BAF60)) [19,20].

With the size of the PPAR ligand binding cavity being significantly larger than that of other nuclear receptors, PPARs succeed in binding numerous natural and synthetic ligands [17,21,22,23,24,25,26,27,28,29,30]. These ligands may trigger an exchange of co-repressors for co-activators that stimulate the function of PPARs [31]. PPARα is highly expressed in the liver, heart, intestine, kidneys, skeletal muscles, and brown adipose tissue, and influences fatty acid metabolism. PPARβ/δ is expressed ubiquitously, and participates in fatty acid oxidation, as well as regulation of blood cholesterol and glucose levels. On the contrary, PPARγ displays highest expression levels in adipocytes, and plays a key role in adipogenesis, lipid biosynthesis, lipoprotein metabolism and insulin sensitivity (Table 1) [16,17].

## 3. The Role of PPARs in Hypertension

PPARs have been reported to play a significant role in the (dys-)regulation of blood pressure.

PPARα is widely known for its pivotal contributions in terms of lipid metabolism, which by extension undoubtedly affects arterial blood pressure [17]. Considering this fact, the PPARα agonists clofibrate and fenofibrate have been shown to effectively reduce blood pressure by interfering with the renin–angiotensin system and consecutively improving the renal function [32,33]. Furthermore, PPARα agonists may inhibit vasoconstriction and induce vasodilatation by either promoting expression of endothelial nitric oxide synthase in endothelium or suppressing agonist-induced endothelin-1 secretion from endothelial cells [34,35]. Of note, PPARα may even mediate transactivation of uncoupling protein 2 or reduction in oxidative stress in specific brain regions responsible for the regulation of blood pressure [36], while PPARα agonist-mediated T cell treatment attenuates secretion of pro-atherogenic cytokines via nuclear factor kappa B (NF-κB) repression induction [37].

Pharmacological PPARβ/δ activation seems to be associated with end-organ damage protection by increasing G protein-coupled signaling regulators, exerting acute nongenomic vasodilator effects, ameliorating the endothelial dysfunction, as well as down-regulating vascular inflammation, vasoconstrictor responses, and sympathetic outflow from the central nervous system [38]. This regulation of both the lipid metabolism and the energy homeostasis might, hence, result in the at least partial regulation of elevated arterial blood pressure, limitation of cardiovascular remodeling and inflammation, nitrogen oxide bioavailability and endothelial function improvement, as well as restriction of oxidative stress and proliferative signaling. Additionally, PPARβ/δ may mediate prostacyclin-induced vasodilatation in renal microvessels, hence contributing to the successful maintenance of renal blood flow [39], whereas PPARβ/δ agonists decrease plasma levels of pro-atherogenic cytokines and autoantibodies [40].

PPARγ is evidently the PPAR isoform with the strongest influence on blood pressure regulation. More precisely, severe hypertension, along with other health conditions including insulin resistance, diabetes, or dyslipidemia, may be partly attributed to the loss of basal transcriptional activity caused by mutations in the wildtype PPARγ [41]. Interestingly, PPARγ has been also suggested to represent a locus interacting at the level of chromatin that could be regarded as a distal associated gene which correlates with blood pressure traits [42]. Furthermore, the PPARγ agonists troglitazone, rosiglitazone, as well as pioglitazone, alone or in combination with different therapeutic agents, may not only improve glycemic control or insulin sensitivity, but also counteract hypertension by lowering blood pressure, as revealed in numerous relevant clinical trials [43,44,45,46]. Recently, Fang et al. published a comprehensive review article on the role of PPARs in hypertension with a special focus on the PPARγ-associated mechanisms of action, and extensively reviewed evidence for its beneficial and very diverse tissue-specific effects in the vasculature, central nervous system, kidney, and immune system on hypertension. In summary, PPARγ activators may regulate blood pressure in the face of volume expansion, interact with specific brain regions directly linked to the short- and long-term regulation of blood pressure, mediate salt/fluid reabsorption, as well as play multiple roles in the regulation of the immune functions involved in the development of hypertension [47].

## 4. The Role of PPARs in Trophoblast Functions and Fetal Origins

All three PPAR isoforms may be expressed in the amnion, decidua, and villous placenta, but are mainly represented in cytotrophoblasts and syncytiotrophoblasts in the first trimester of the placenta, signifying their exclusive contributions to trophoblast differentiation and functions of the placenta [48]. PPARβ plays an essential role in embryonic implantation and placentation by promoting the differentiation of trophoblast giant cells through the activation of the phosphatidylinositol 3-kinase (PI3K)/protein kinase B (AKT) signaling pathway, as well as the regulation of the expression of transcription factor I-MFA [49]. With the chorion-specific transcription factor-1 (GCM-1) and the chorionic gonadotropin beta-subunit (hCGβ) being the biochemical markers of trophoblast differentiation [50], PPARγ increases hCGβ secretion in the first- and third-trimester primary villous trophoblasts and, consecutively, indirectly stimulates cytotrophoblast differentiation into syncytiotrophoblasts [51,52]. Notably, PPARγ evidently activates the increase in the hCG α- and β-subunit transcript levels in villous cytotrophoblasts, but decreases their expression levels in extra-villous trophoblasts, respectively [53]. During the trophoblast stem cell differentiation, PPARγ even targets different genes and is diversely expressed in different trophoblast subsets and times [54]. Furthermore, microRNA-1246 has been shown to up-regulate the expression of PPARγ and induce syncytiotrophoblast differentiation through the inhibition of the WNT/β-catenin signaling pathway [55], while hypoxia has been suggested to have a negative effect on the expression of PPARγ with the inactivation of both hypoxia inducible factor (HIF) and histone deacetylases (HDACs), which hinders the trophoblast differentiation [56]. Collectively, PPARγ seems to both spatially and temporally affect any subtype of trophoblast lineage differentiation during pregnancy.

In terms of trophoblast secretion, PPARγ signaling pathways have been described to promote the energy metabolism of trophoblast cells through the secretion of inflammatory cytokines, including interferon (IFN)-γ and prostaglandin E2 (PGE2) [57]. Both PPARβ and PPARγ are also associated with trophoblast cell fusion and syncytiotrophoblast formation via direct modulation of the target gene syncytin-1 [58,59]. In the context of trophoblast invasion, the molecular mechanism of PPARγ may efficiently decrease pregnancy-associated plasma protein-A (PAPP-A) and avert the secretion of insulin-like growth factor (IFGII) [60].

Moreover, PPARs are involved in the modulation of trophoblastic fat transport, fat storage, and fat metabolism by up-regulating the lipid droplet that is associated with the protein adipophilin [61]. Importantly, PPARγ is necessary for adipogenesis and normal insulin sensitivity. Hyperglycemia enhances PPARγ pathways, thus diminishing the human cytotrophoblast invasion, increasing interleukin-6 and soluble fms-like tyrosine kinase-1 (sFIt-1), as well as inhibiting urokinase plasminogen activator (uPA) and plasminogen activator inhibitor 1 (PAI-1) [62].

All three PPAR isoforms have been reported to be expressed in cells of the endoderm and mesoderm at early time points during the development of the human embryo and fetus [63], with PPARα and PPARγ exhibiting first placental and then fetal expression [64]. As with adult tissues, PPARα plays a significant role in lipid catabolism in the fetal liver and heart [65,66]. PPARβ/δ is expressed in the early stages of organogenesis [67] and effectively contributes to the closure of the neural tube [68]. PPARγ seems to have potent anti-inflammatory effects by preventing the overproduction of both nitric oxide and matrix metalloproteinases in fetuses from diabetic rats [69]. Recently, Barbieri et al. studied the association of selected PPARγ single-nucleotide polymorphisms with intrauterine growth restriction through an unmatched case-control study nested in a prospective cohort study in a cohort of live births in Ribeirão Preto. Surprisingly, even after controlling for non-genetic confounding factors, the PPARγ single-nucleotide polymorphism rs41516544, which has been associated with type 2 diabetes with an elevated risk of metabolic complications (e.g., insulin resistance, obesity, etc.), was suggested to augment the risk of intrauterine growth restriction for both male and female offspring [70]. Altogether, PPAR methylation pattern alterations during early development may manifest in fetuses and be maintained throughout the life course and even across generations [71].

## 5. The Role of PPARs in Uterine and Placental Angiogenesis

PPARs have been suggested to exhibit an interplay between endothelial aging and angiogenesis [72]. PPARα triggers the caspase pathway by up-regulating thrombospondin-1 (TSP-1) expression, and, therefore, results in apoptosis of endothelial cells, angiogenesis inhibition, as well as downregulation of tumor progression. Gypenoside 140 (GP-140) up-regulation results in reduced promoter luciferase secretion of the *vascular endothelial growth factor* (*VEGF*) gene, which moderates the process of angiogenesis, respectively [73]. Moreover, PPARα has an inhibitory effect on vascular smooth muscle proliferation through the activation of the cyclin-dependent kinase inhibitor p16(INK4a) which results in cell cycle arrest and reduced endothelial proliferation [74]. In the case of PPARβ/δ, the angiogenic profile of endothelial progenitor cells is directly related to the activity of this PPAR isoform, as well as the activation of matrix metalloproteinases [75,76]. In terms of PPARγ contributions, the PPARγ agonist rosiglitazone was applied in wild-type mice during later stages of embryonic development, hence disorganizing the placental layers, altering placental microvasculature, as well as leading to a decreased expression of proangiogenic genes ranging from *Prl2c2* or *VEGF* to *Platelet and Endothelial Cell Adhesion Molecule 1* (*Pecam1*) [77]. Similarly, the partial agonist of PPARγ and angiotensin-II receptor antagonist telmisartan has been described to reduce functional microvessel density and blood perfusion, down-regulate immune cell infiltration and cell proliferation, and correlate with the pathogenesis of smaller endometriotic lesion sizes [78]. Of note, Zhang et al. recently investigated the expression of PPARγ in porcine uteroplacental tissues at three different gestational days, and employed both a cell model of porcine umbilical vein endothelial cells and a mouse xenograft model to assess angiogenesis in vitro and in vivo. The main PPARγ locations were described to be in the glandular epithelium, trophoblast, amniotic chorion epithelium and vascular endothelium, while PPARγ expression was determined to exhibit significant down-regulation in placenta with dead fetus. In porcine umbilical vein endothelial cells, PPARγ knock-out significantly repressed proliferation, migration and tube formation in vitro. In vivo, capillary formation in mouse xenograft models was hindered by blocking S-phase, promoting apoptosis and down-regulating the VEGF angiogenic factors along with its receptors [79].

## 6. The Role of PPARα in Preeclampsia

Several studies have investigated the role of PPARα in preeclampsia.

Both Chang et al. and Dekker Nitert et al. examined PPARα levels in human placentae, and discovered a significantly lower PPARα mRNA and protein expression in preeclamptic placentae [80,81]. On the contrary, He et al. reported significantly higher PPARα mRNA and protein expression levels that inversely correlated with the 11β-HSD2 protein level in preeclamptic placentae [82], while Holdsworth-Carson et al. showed that placental PPARα protein was significantly increased in placentae from women that delivered preterm with co-existing intrauterine growth restriction and preeclampsia [83].

Santana-Garrido et al. obtained placenta samples from normal pregnant and preeclamptic rats, and described significant depletion for PPARα gene/protein expression in preeclampsia [84].

## 7. The Role of PPARβ/δ in Preeclampsia

Rodie et al. performed immunohistochemistry in third trimester preeclamptic placentae and reported that PPARδ was localized to the syncytium and cells within the stroma. Of note, there was no difference in PPARδ mRNA and protein expression between preeclamptic placentae and placentae from uncomplicated pregnancies [85]. On the other hand, He et al. reported significantly higher PPARβ mRNA and protein expression levels in preeclamptic placentae [82].

## 8. The Role of PPARγ in Preeclampsia

PPARγ is undoubtedly the most studied PPAR in preeclampsia.

A great number of studies has explored the role of PPARγ in preeclamptic human placentae. Different study groups reported that PPARγ seems to be down-regulated in preeclamptic placentae [58,82,86,87,88,89,90,91,92,93,94]. Of note, placental 11β-hydroxysteroid dehydrogenase type 2 (11β-HSD2) and specificity protein 1 (Sp-1) protein levels were positively associated with PPARγ [82], omega-3 fatty acid supplementation promoted PPARγ protein expression in late-onset preeclampsia [88], PPARγ expression positively correlated with the epigenetic modification via trimethylated lysine 4 of the histone H3 (H3K4me3) and acetylated lysine 9 of the histone H3 (H3K9ac) [91], while PPARγ was negatively associated with micro-RNA-27b-3p and showed good predictive value on preeclampsia [94]. Hastie et al. determined the mRNA expression of the PPARγ co-activator 1 (PGC1α) to be also reduced in preeclamptic placentae [95]. On the contrary, Holdsworth-Carson et al. showed that placental PPARγ mRNA and protein were significantly increased in placentae from women that delivered preterm with co-existing intrauterine growth restriction and preeclampsia [83], while Jia et al. noted high PPARγ protein expression levels in preeclamptic placental tissues, with the rs201018 polymorphism in the *PPARγ* promoter region significantly correlating with the development of preeclampsia [96]. Liu et al. showed that the C1431T variant of PPARγ significantly correlates with preeclamptic susceptibility [97]. In addition, Permadi et al. exposed primary trophoblastic cells obtained from normal pregnancy to serum from normal pregnancy, early-onset, or late-onset preeclampsia, and highlighted that exposure to serum from only late-onset preeclampsia induced high PPARγ expression [98]. Certain research groups could, however, not detect any correlation between PPARγ (polymorphism) and the occurrence of preeclampsia [81,85,99], with the clinical course of preeclampsia not differing among PPARγ genotypes [99]. Ghorbani et al. studied the gene variants in preeclamptic and healthy pregnant women using polymerase chain reaction–restriction fragment length polymorphism (PCR-RFLP), and underlined the missing correlation of the gene variants of PPARγ Pro12Ala and RXR-α with the risk of preeclampsia [100].

Armistead et al. cultured preeclamptic placentae that displayed decreased PPARγ protein expression. After administration of rosiglitazone, both PPARγ and Glial Cell Missing 1 (GCM1) expression were restored, while sFIt-1 expression was down-regulated [101]. Additionally, Kohan-Ghadr et al. incubated term preeclamptic placental explants with rosiglitazone, which induced Reactive Oxygen Species (ROS) scavenger expression [102]. Similarly, Luminex assay results supported the notion that rosiglitazone-induced PPARγ activation in preeclamptic placentae, on the one hand, significantly reduces Angiopoeitin-2 (Ang-2) and soluble Endoglin (sEng), but, on the other hand, increases placental growth factor (PIGF), fibroblast growth factor-2 (FGF-2), heparin-binding epidermal growth factor (HB-EGF), and follistatin (FST) protein secretion, hence improving the overall placental angiogenic profile [103]. Furthermore, Liu et al. investigated the effect of a bioactive food compound on preeclampsia in placental explant cultures and RUPP rat model, and demonstrated that procyanidin B2 induced PPARγ expression, thus inhibiting sFLT1 expression via the nuclear factor erythroid 2-related factor 2 (Nrf2)/PPARγ axis [104].

Different study groups have examined the role of PPARγ in preeclamptic pregnant animal models as well. After obtaining placenta samples from normal pregnant and preeclamptic rats, Santana-Garrido et al. described significant PPARγ gene/protein expression depletion in preeclampsia [84]. El-Saka et al. experimentally induced preeclampsia in Wistar rats, and described a significant PPARγ expression decrease that significantly improved after angiotensin 1–7 treatment [105]. Moreover, Nema et al. showed that placental PPARγ levels were significantly lower in Vitamin D-deficient than in Vitamin D-supplemented female rats with preeclampsia [106], while Ueki et al. concluded that plasma from Catechol-O-Methyltransferase (COMT)-deficient pregnant mice exhibited low PPARγ transcriptional activity, and correlated with elevated angiotensin II receptor type 1 (AT1R) levels [107].

Interestingly, some research groups also chose to study the effects of rosiglitazone, a PPARγ agonist, in preeclamptic animal models. Allam et al. treated preeclamptic pregnant rats with rosiglitazone from days 14 to 20, and reported blood pressure reduction, renal function amelioration, endothelin-1, angiotensin-II, and interleukin-6 decrease, nitric oxide level increase, as well as fetal weight gain [108]. Furthermore, McCarthy et al. simultaneously treated reduced uterine perfusion pressure (RUPP) preeclamptic rats with rosiglitazone and the heme oxygenase 1 inhibitor tinprotoporphyrin IX, and proved their synergistic positive effects on hypertension and endothelial function [109]. In an attempt to further endorse their assertion, the same study group administered the PPARγ antagonist T0070907 to healthy pregnant rats, and determined that the inhibition of PPARγ-associated pathways resulted in hypertension, intrauterine growth restriction, proteinuria, vascular dysfunction, as well as elevated platelet aggregation [110]. Notably, Zhang et al. employed Sprague–Dawley rats, and, after administration of different doses of aspirin in presence or absence of T0070907, suggested that aspirin potently reverses the pro-preeclamptic effects of the PPARγ antagonist [111]. Similarly, Guo et al. administered low-dose aspirin to a PPARγ-antagonist-treated mouse model of preeclampsia, and reported effective blood pressure and urinary protein level decrease, endoglin and interleukin-β inhibition, as well as placental lesion alleviation and weight increase [112]. Moreover, Holobotovskyy et al. employed Regulator of G protein Signaling 5 (RGS5) heterozygote knockout mice, and treated them with the PPARγ agonist troglitazone, thus achieving vascular function, blood pressure, as well as birth weight normalization [113].

Table 2 summarizes the role of PPARs in preeclampsia.

Figure 3 depicts the effects of PPARγ-modulating treatment agents in preeclampsia.

## 9. Conclusions

To date, several review articles have been published on the role of PPARs in trophoblast differentiation and healthy placental development [13,14,114]. However, these publications mainly focus on the metabolic regulation of PPARs affecting trophoblast physiological conditions, but do not deeply investigate the influence of PPAR dysfunctions in trophoblasts on pregnancy diseases. Notably, the present article represents the most comprehensive up-to-date literature review incorporating a total of 35 original research articles on the role of PPARs in preeclampsia, a potentially life-threatening risk factor for disturbed pregnancy. Not only does it provide a detailed insight into the PPAR-associated preeclamptic pathogenesis, but it also examines the therapeutic effect of PPAR-modulating drugs in preeclampsia.

PPARs seem to play a significant role in the pathophysiology of preeclampsia. Compared to PPARγ, only a handful of studies have reported on the expression of PPARα and PPARβ/δ in preeclamptic placentae. Despite the limited number of available publications, study results are contradictory, thus not allowing for a conclusive statement on the up- or down-regulation of these receptors in preeclampsia yet.

On the contrary, a significantly greater number of studies have investigated the role of PPARγ in preeclampsia, with the majority of these studies reporting a reduction in PPARγ in preeclampsia. Nevertheless, several study groups did not detect differences between healthy and preeclamptic placentae, while some authors even suggested PPARγ up-regulation in preeclampsia. A possible explanation for this discrepancy could be the lack of large-scale studies incorporating numerous study samples that would allow for reliable and statistically robust observations.

Importantly, several research groups have evaluated the effects of PPARγ agonists alone, or in combination with other pharmaceutical agents, in preeclamptic placental explant cultures and mouse models, and underlined their promising therapeutic effects. Among them, rosiglitazone, an insulin sensitizer in the thiazolidinedione class, represents the most studied treatment agent, which, except for its mighty antidiabetic effects, could potentially revolutionize anti-preeclamptic therapy. Consequently, the verification of these findings in large-scale clinical trials might lay the basis for groundbreaking novel treatment alternatives for preeclamptic pregnancies.

In summary, this review highlights the crucial role of PPARs in preeclampsia and paves the way for future research directions.

## Figures and Tables

**Figure 1 cells-12-00647-f001:**
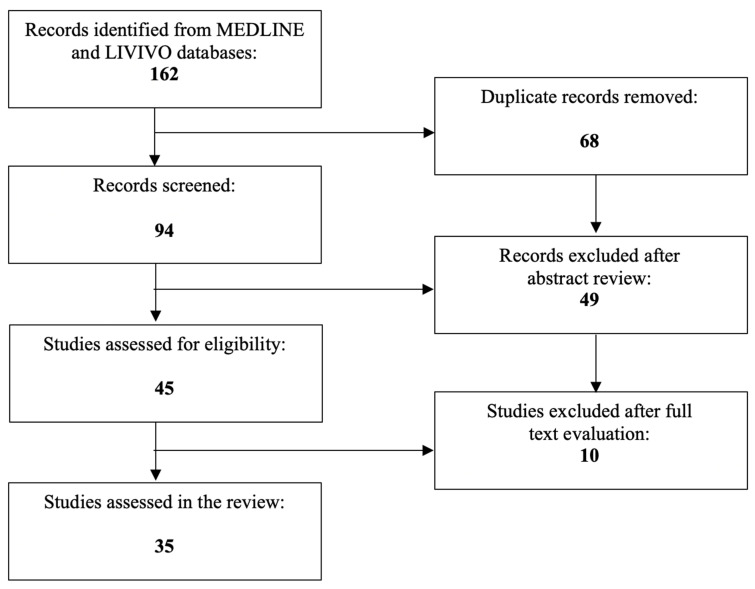
PRISMA flow diagram visually summarizing the screening process.

**Figure 2 cells-12-00647-f002:**
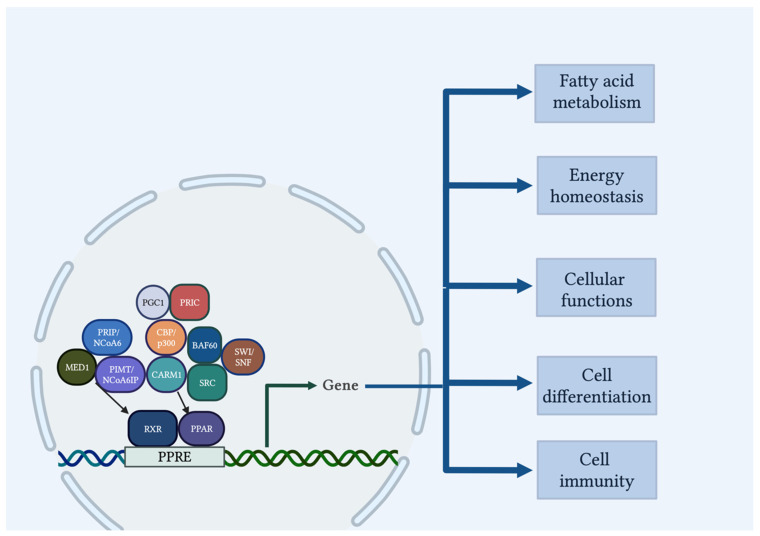
PPAR signaling pathway. Upon activation of PPARs and ligands, a heterodimer with RXR is formed, which binds to the PPRE upstream of the target gene promoter, hence regulating the transcription of target genes responsible for energy production, lipid metabolism, cellular functions, and inflammation. Co-activators influence the functioning of many regulators and affect gene transcription. Created with BioRender.com, accessed on 27 December 2022.

**Figure 3 cells-12-00647-f003:**
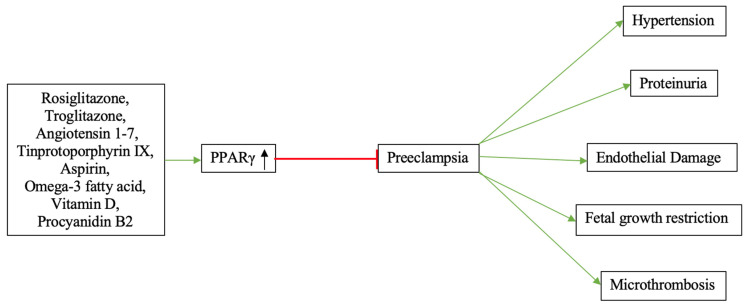
The effects of PPARγ-modulating treatment agents in preeclampsia. By inducing PPARγ activation, diverse treatment agents may improve the overall placental angiogenic profile, thus exhibiting anti-preeclamptic effects.

**Table 1 cells-12-00647-t001:** Expression sites, functions and ligands of PPARs.

PPARs	Expression Sites	Functions	Natural Ligands	Synthetic Ligands
PPARα	Liver, Heart, Intestine, Kidneys, Skeletal muscles, Brown adipose tissue	Fatty acid metabolism	Unsaturated fatty acids Leukotrienes Eicosatetraenoic acids	Fibrates
PPARβ/δ	Ubiquitously	Fatty acid oxidation, Blood cholesterol and glucose level regulation	Unsaturated fatty acids Prostacyclins Very low-density lipoproteins	GW501516
PPARγ	Adipocytes	Adipogenesis, Lipid biosynthesis, Lipoprotein metabolism, Insulin sensitivity	Unsaturated fatty acids Eicosatetraenoic acids Octadecadienoic acids Prostaglandins	(Non-)Thiazolidinedioneinsulin sensitizers Angiotensin II receptor antagonists Nonsteroidal anti-inflammatory drugs Synthetic cannabinoids

**Table 2 cells-12-00647-t002:** The role of PPARs in preeclampsia.

PPARs	Models	Expression Levels	Treatment Agents	References
PPARα	Rat, Human	↓	None	[80,81,84]
	Human	↑		[82,83]
PPARβ/δ	Human	↔	None	[85]
	Human	↑		[82]
PPARγ	Mouse, Rat, Human	↓	Rosiglitazone, Troglitazone, Angiotensin 1–7, Tinprotoporphyrin IX, Aspirin, Omega-3 fatty acid, Vitamin D, Procyanidin B2	[58,82,84,86,87,88,89,90,91,92,93,94,101,102,103,104,105,106,107,108,109,110,111,112,113]
	Human	↑	None	[83,96,98]
	Human	↔		[81,85,99,100]

## Data Availability

Not applicable.

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
