# Peer review of "The Role of Peroxisome Proliferator-Activated Receptors in Preeclampsia"

_cells, 2023, doi:10.3390/cells12040647_

Round 1

Reviewer 1 Report

Psilopatis et al. reviewed 35 research articles reporting the role of PPARs (Peroxisome Proliferator-Activated Receptors) in preeclampsia (PE) and summarised their relevant abundances at the mRNA/protein level and chemical substances that could potentially regulate their abundance in the presence of PPAR-antagonists or pathological conditions. In general, the manuscript reads well and the authors nicely summarise the role of PPARs in PE using tables and a figure. This reviewer does not have any major comment (but some minor points shown below) nor any objection against publication.

Minor comments:

1. Some of the legends in Figure 1 are not complete. For example, CBP/p300, and P-γC1 were neither annotated nor mentioned in the manuscript. Also, the size of a cell nuclear (NB, not annotated) could be reduced and the font size of gene functions (i.e. those in the boxes in the far-right) could be increased to enhance readability.  

2. Having said the above, the authors might want to consider adding a description of PPAR-coactivators in Section 2 (PPAR structure and mechanism of action) and their potential role in PE. The paper from Yu and Reddy 2007 (pmid: 17306620) could be helpful. The authors could appreciate that there are far more PPAR-coactivators that could be added in Figure 1.

3. Line 168-173 – this sentence is too long and complex, so it is recommended to rewrite. 

Reviewer 2 Report

In this review, Psilopatis and colleagues, provide general background on PPARs and their overall metabolic effects and locations.  They then review human and animal studies that have evaluated increases, decreases or no changes in the different PPARs in preeclampsia or models.  They clearly indicate how they chose the studies being reviewed and provide their search strategy.  In general, excellent information is provided.  Overall, the figures and tables could be better designed to make the material easier to digest for the reader.  Putting the human data first and then commenting on the value of the animal studies in modeling what is happening in the human would be a valuable effort.

Concerns:

1.  Table 1:  An expansion of Table 1 could enhance the manuscript and include the information in lines 79-84 as relates to both the sites of expression as well as the major functions of the different receptors.  This is appropriate if the table does not get too large with this change (or add another table).

2.  Consider presenting the human data first in each section, followed by the animal data.

3.  In the PPAR gamma section, the studies should be clear as to whether they use rats and mice.  On line 105 they indicate mouse models and then they immediately discuss rat models.  As they discuss the preeclampsia models, they should define which model is being used rather than have the reader search out the reference.  A more detailed table (an expanded Table 2) with the PPAR gamma studies would be valuable, especially if it outlined the model and species (human vs rat vs mouse) used.  Such a detailed outline might reveal some trends for situations in which PPAR gamma is not changed or upregulated.  The role of PPAR gamma could certainly differ depending on the preeclampsia model and this is seemingly not considered. 

4.  Figure 1:  The figure legend should define each of the subunits represented in the Figure without having to refer to the text.  In the text they refer to PPREs and if possible, including this on the diagram would be advantageous.  If the different components are indicated in the figure, they should be addressed in the text.  More background on PPAR signaling would be useful to appreciate the studies cited in the later sections.

5.  Figure 2:  The overall message from Figure 2 is not totally clear and would profit from more legend or more discussion in the text.  The Figure does not stand on its own well. It is unclear if the authors are saying that all of the treatments listed will decrease PPAR gamma, or that once PPAR gamma is decreased then all of these agents will stop preeclampsia from developing.  The legend suggests that all of these agents activate PPAR gamma and thus counteract the PPAR gamma downregulation but this is not clearly depicted in the figure.

6.  The general message is that PPAR gamma downregulation is accepted.  However, the authors should articulate how they arrived at that overall conclusion given the 7 studies that report no change or an increase. 

Round 2

Reviewer 2 Report

The authors were responsive to the review and have expanded and improved the manuscript and figures with this revision.